# Matrix Metalloproteinases and Their Role in Mechanisms Underlying Effects of Quercetin on Heart Function in Aged Zucker Diabetic Fatty Rats

**DOI:** 10.3390/ijms22094457

**Published:** 2021-04-24

**Authors:** Barbora Boťanská, Monika Barteková, Kristína Ferenczyová, Mária Fogarassyová, Lucia Kindernay, Miroslav Barančík

**Affiliations:** Centre of Experimental Medicine, Institute for Heart Research, Slovak Academy of Sciences, Dúbravská Cesta 9, 841 04 Bratislava, Slovakia; barbora.botanska@gmail.com (B.B.); Monika.Bartekova@savba.sk (M.B.); kristina.ferenczyova@savba.sk (K.F.); Fogarassy.alexander@gmail.com (M.F.); lucia.griecsova@savba.sk (L.K.)

**Keywords:** matrix metalloproteinases, quercetin, Zucker Diabetic Fatty rats, heart, cell signaling

## Abstract

Several mechanisms may contribute to cardiovascular pathology associated with diabetes, including dysregulation of matrix metalloproteinases (MMPs). Quercetin (QCT) is a substance with preventive effects in treatment of cardiovascular diseases and diabetes. The aim of the present study was to explore effects of chronic QCT administration on changes in heart function in aged lean and obese Zucker Diabetic Fatty (ZDF) rats and that in association with MMPs. Signaling underlying effects of diabetes and QCT were also investigated. In the study, we used one-year-old lean and obese ZDF rats treated for 6 weeks with QCT. Results showed that obesity worsened heart function and this was associated with MMP-2 upregulation, MMP-28 downregulation, and inhibition of superoxide dismutases (SODs). Treatment with QCT did not modulate diabetes-induced changes in heart function and MMPs. However, QCT activated Akt kinase and reversed effects of diabetes on SODs inhibition. In conclusion, worsened heart function due to obesity involved changes in MMP-2 and MMP-28 and attenuation of antioxidant defense by SOD. QCT did not have positive effects on improvement of heart function or modulation of MMPs. Nevertheless, its application mediated activation of adaptive responses against oxidative stress through Akt kinase and prevention of diabetes-induced negative effects on antioxidant defense by SODs.

## 1. Introduction

Obesity is a serious health problem and an important factor contributing to the development of metabolic syndrome. This is associated with increased risk of type 2 diabetes [1] and cardiovascular disease development [2]. Moreover, cardiovascular complications significantly increase morbidity and mortality triggered by diabetes mellitus, which substantially increases the risk of ischemia/reperfusion-induced injury, myocardial infarction, and heart failure [3,4,5]. Several mechanisms may contribute to cardiovascular pathology associated with diabetes, including remodeling and disturbed turnover of extracellular matrix (ECM). ECM is a dynamic structure that requires constant turnover, synthesis, and degradation of its components. Matrix metalloproteinases (MMPs) are enzymes playing an important role in the degradation of ECM proteins through their cleavage. The function of MMPs is tightly controlled by tissue inhibitors of matrix metalloproteinases (TIMPs) and dysregulation of MMPs and/or TIMPs is involved in pathology development [6,7,8].

Previous reports indicated that MMPs are involved in several forms of cardiac disease, including ischemia-reperfusion (I/R) injury, post-infarction left ventricular remodeling, heart failure, and dilated cardiomyopathy [9,10,11]. Elevation of circulating plasma MMPs was observed after acute myocardial infarction (MI) [12,13] as well as in consequence of cardiotoxic effects of doxorubicin [14,15]. Several lines of evidence also point to a close association between cardiac MMPs and diabetes. MMP-2 and MMP-9 have been shown to contribute to cardiac fibrosis in diabetes [16,17] and diabetes was found to be enhanced through increased oxidative stress vascular MMP-9 activity [18,19].

Various polyphenols were studied for their effects on modulation of diabetic complications [20,21,22]. One of them is quercetin (QCT), a polyphenolic compound present in various foods (vegetables, fruit, and wine). This flavonoid possesses various biological functions, including antioxidative, anti-inflammatory, and anticoagulation activities [23,24,25]. QCT has also been found to have preventive effects in the treatment of cardiovascular diseases and diabetes. Treatment with QCT exerted antihypertensive effects [26], as well as cardioprotective effects against ischemia-reperfusion (I/R) injury [27,28] or doxorubicin-induced cardiotoxicity [14]. QCT has also been reported to protect through antioxidant mechanisms against myocardial infarction in streptozotocin (STZ)-induced diabetic rats [20]. Moreover, application of QCT reduced blood glucose levels in STZ-induced [29] and in type 2 diabetic rats [21].

The aim of this study was to explore the effects of prolonged administration of quercetin on the changes induced by obesity and diabetes development in one-year-old Zucker Diabetic Fatty (ZDF) rats, the animal model of type 2 diabetes. The purpose was to assess the extent to which prolonged quercetin administration alters heart function in obese diabetic ZDF rats and that in addition to matrix metalloproteinases as markers of tissue remodeling. The potential molecular mechanisms underlying the actions of QCT in aged lean and obese ZDF rats were also investigated.

## 2. Results

### 2.1. Effect of Quercetin on Biometric and Biochemical Characteristics of Lean and Obese ZDF Rats

Increased obesity and development of diabetes in one-year-old diabetic ZDF rats were confirmed by significant increase of body weight and levels of fasting glucose (glycemia) in these type 2 diabetic animals (Table 1). In consequence of diabetes development, there were significantly increased plasma levels of total cholesterol, triglycerides, low-density lipoprotein (LDL)-cholesterol, and high-density lipoprotein (HDL)-cholesterol. Treatment with QCT had no significant effects on these parameters in both control lean and obese diabetic rats (Table 1).

### 2.2. Effects of Quercetin on Functional Parameters of Isolated Langendorff-Perfused Hearts

The effects of diabetes and quercetin treatment on heart function were examined by assessing hemodynamic parameters of isolated perfused rat hearts. We demonstrated significant effect of diabetes on most of the hemodynamic parameters of the hearts. We observed that diabetes caused a significant decrease in heart rate and increase in parameters left ventricle developed pressure (LVDP), maximal rates of pressure development/fall (+(dp/dt)_max_, −(dp/dt)_max_), and coronary flow (Figure 1). These effects of diabetes were not influenced by quercetin treatment. Quercetin also did not influence heart functional parameters in control lean ZDF rats.

### 2.3. Effects of Quercetin on Modulation of Matrix Metalloproteinase-2 Activities and Protein Levels in Lean and Obese ZDF Rats

The MMPs activities in left ventricular heart tissue samples were analyzed by zymography using gelatin as a substrate. The positions of individual MMPs were identified using corresponding positive controls. We found that in obese diabetic rats, there were significantly upregulated activities of 72-kDa form of MMP-2 (Figure 2A,D). The application of quercetin did not influence these diabetes-induced effects on cardiac MMP-2 activation. The observed effects of diabetes development on 72 kDa MMP-2 activities were not associated with a modulation of the protein levels of this enzyme (Figure 2B,D). In obese diabetic ZDF rats, there were increased protein levels of 63 kDa MMP-2 and treatment with quercetin prevented the diabetes-induced changes. However, the changes in protein levels of this form of MMP-2 were not followed by modulation of activities (Figure 2C,D).

### 2.4. Effects of Quercetin on Matrix Metalloproteinase-9 and Tissue Inhibitor of Matrix Metalloproteinases-2 in Lean and Obese ZDF Rats

In contrast to 72 kDa MMP-2, by using gelatin zymography we did not detect changes in activities of MMP-9 in samples prepared from left ventricular tissue. Western blot analysis of protein levels of this enzyme showed that diabetes development was not associated with a modulation of its levels (Figure 3A,C). We also did not observe significant effects of quercetin treatment on MMP-9 protein levels in lean and obese diabetic ZDF rats. The functions of MMP-2 and MMP-9 are tightly controlled by tissue inhibitors of matrix metalloproteinases (TIMPs). We analyzed the effects of quercetin on TIMP-2, but we did not find influence of this flavonoid on TIMP-2 in both control lean and obese diabetic rats (Figure 3B,C)

### 2.5. Effects of Quercetin on Matrix Metalloproteinase-28 in Lean and Obese ZDF Rats

Matrix metalloproteinase-28 (MMP-28) is the newest identified member of the MMP family. Our results showed a downregulation of protein levels of this enzyme in obese diabetic ZDF rats (Figure 4A,B). The application of quercetin did not change MMP-28 protein levels in control lean animals, and also did not influence the diabetes-induced effects on cardiac MMP-28.

### 2.6. Effects of Quercetin on Collagen I Content in Lean and Obese ZDF Rats

Detection with a specific antibody documented decreased content of collagen I in the left ventricle of obese diabetic ZDF rats (Figure 5A,B). Collagen I is a potential endogenous substrate for MMPs, and the diabetes-induced changes may reflect the observed activation of tissue 72 kDa MMP-2. The observed data show that quercetin did not change collagen I levels in control lean rats and also did not prevent the effects of diabetes (Figure 5A,B).

### 2.7. Quercetin Prevents Diabetes-induced Inhibition of Superoxide Dismutase

The activation of the non-cleaved, oxidatively activated 72-kDa form of tissue ventricular MMP-2 in diabetic hearts (Figure 2A,D) suggested potential alterations in activities of enzymes involved in radical formation. We found that the effects of diabetes were associated with reduction of total superoxide dismutase (SOD) activities and QCT treatment prevented the negative effects of diabetes on SOD inhibition (Figure 6C). The observed changes in myocardial SOD activities in obese ZDF rats were in positive correlation with protein levels of SOD-1 and SOD-2 isoforms. Diabetes induced downregulation of both SOD-1 and SOD-2 protein (Figure 6A,B,D). Quercetin reversed the effects of diabetes on SOD-2 and induced significant increase of SOD-2 levels in comparison to diabetic vehicle-treated hearts (Figure 6B,D). On the other hand, the effects of QCT on the protein expression of SOD-1 were not significant (Figure 6A,D).

### 2.8. Quercetin Induced Akt Kinase Activation in Diabetic Rat Hearts

To further characterize the possible mechanisms involved in effects of diabetes and QCT, we investigated the changes in Akt kinase. The obtained data showed that there are no differences in the levels of total Akt kinase between the experimental groups (Figure 7B). However, detection with a phosphospecific antibody revealed an increased phosphorylation of Akt kinase specifically on Ser473 in the tissue of the left ventricle of rat hearts exposed to the effects of QCT, especially in obese diabetic ZDF rats (Figure 7B). The observed levels of active Ser473 phosphorylated Akt kinase after QCT treatment were also increased in relation to total Akt kinase levels (Figure 7A).

## 3. Discussion

The effects of diabetes and quercetin treatment on heart function were examined by assessing hemodynamic parameters of isolated perfused rat hearts. The results of the present study clearly show significant influence of increased obesity on hemodynamic parameters of the hearts in year-old diabetic ZDF rats. We observed that diabetes caused a significant decrease in heart rate and increase in parameters LVDP, +(dp/dt)_max_, −(dp/dt)_max_, and coronary flow. On the other hand, we found no effect of quercetin treatment on these heart functional parameters. Changes observed in obese diabetic ZDF rats are in accordance with several studies that demonstrated decreased heart rate in ZDF rats in vivo [30,31] as well as in STZ-induced diabetic rat hearts [32]. On the contrary, our results are not in line with observations of diabetes-induced changes in contractile parameters in several studies since no significant changes in LVDP were documented in ZDF rats [33], and parameters +(dp/dt)_max_ and −(dp/dt)_max_ were even found to be decreased in STZ-induced diabetic rat hearts [32]. It should be noted that the rats in the abovementioned studies were of a different age as compared to our study, thus age might be a potentially important factor that influences hemodynamics in type 2 diabetic hearts. However, there are in fact no studies demonstrating the effect of age on hemodynamic parameters in ZDF rats. Only the study of Wang and Chatham [34] revealed the effect of age on hemodynamic parameters of isolated ZDF rat hearts; however, the oldest group in this study was 6 months old, which is still significantly younger than the rats used in our study. They demonstrated that LVDP did not change, while +(dp/dt)_max_ and −(dp/dt)_max_ were decreased in the diabetic group compared to the control at this age. Thus, it can only be speculated that increased contractility in diabetic hearts observed in our study might be due to heart hypertrophy (increased heart weight) found in aged ZDF rats as compared to their age-matched controls. Increase in heart weight might also be in tight relation to the increased levels of coronary flow in diabetic hearts as compared to non-diabetic controls documented in the current study.

By the study of molecular mechanisms underlying the effects of obesity development and possible actions of QCT in aged lean and obese ZDF rats we observed the potential role of MMP-2 and MMP-28. Interestingly, we found different effects of diabetes on distinct MMPs. MMP-2 was upregulated, MMP-9 was unchanged, and MMP-28 was downregulated in consequence of diabetes development. MMPs and their tissue inhibitors play a key role in regulation of extracellular matrix (ECM) remodeling. Changes in MMPs activation could be related to structural disorganizations of the cardiac extracellular space associated with disruption of cardiac function. Increased MMPs were found to be related to development of cardiac fibrosis and left ventricular diastolic dysfunction [35]. Moreover, application of MMP inhibitor was found to protect the heart against ischemia/reperfusion and this was associated with improved recovery of cardiac work, cardiac output, and aortic flow during reperfusion [36]. We found in the left ventricle of diabetic ZDF rats increased activities of 72 kDa MMP-2. MMP-2 is an enzyme that plays an important role in processes of extracellular matrix remodeling and its activation could be related to structural disorganizations of the cardiac extracellular space. Several studies have documented the role of this enzyme during diabetes. However, the data point to potentially different roles of MMP-2 depending on conditions of diabetes. Similar to our findings, hyperglycemic stimuli in both in vitro and in vivo diabetic cardiac models led to induction of MMP-2 (length and N-terminal truncated isoforms) [10]. On the other hand, in porcine model of STZ-induced diabetes, decreased cardiac MMP-2 activities in comparison to non-diabetic animals were demonstrated [37]. Changes in activities of 72 kDa form of MMP-2 observed in our study were not associated with modulation of 72 kDa MMP-2 protein levels. This form of MMP-2 can be activated through conformational changes induced by radicals such as superoxide. This suggests a potential role of alterations in enzyme activities of SOD, enzyme involved in radical superoxide formation. We found that activation of 72-kDa MMP-2 in diabetic hearts was associated with reduction of total SOD activities as well as protein levels of soluble cytoplasmic SOD-1 and mitochondrial SOD-2. Thus, the observed effects of diabetes development on changes in tissue 72-kDa MMP-2 activities can also be explained through the observed effects on SOD expression and activities. Treatment with QCT prevented the negative effects of obesity on SOD but did not modulate the diabetes-induced changes in 72 kDa MMP-2 activation. These findings are different from our previous observations documenting efficiency of QCT in reversal of negative effects of DOX on both MMP-2 activation and SOD inhibition in non-diabetic animals [14].

Worsening of heart function observed in obese ZDF rats could also be associated with changes in serum uric acid (UA) levels. UA has both pro-oxidant and antioxidant activity and is a biomarker of heart health risk. Its overproduction can trigger oxidative stress associated with increased reactive oxygen species production, lipid peroxidation, and reduced nitric oxide availability in endothelial cells [38]. Higher serum UA concentrations were found to be associated with higher mean arterial pressure and hypertension [39] as well as with an increased risk of type 2 diabetes [40]. A possible positive relation of serum uric acid to MMPs [41,42] as well as negative correlation between SOD and uric acid were documented. It has been found that decreased levels of antioxidants such as SOD in plasma and tissues are associated with increased UA levels in blood circulation [43]. Our finding that supplementation with QCT prevented the negative effects of diabetes on SOD indicate that some actions of QCT could also be realized through modulation of serum UA concentrations associated with reduction of oxidative stress. This is also supported by the finding that QCT improved oxidative status in rats exposed to isoproterenol-induced myocardial damage by reversal of negative effects of isoproterenol (ISO) on myocardial SOD inhibition and decrease of ISO-induced upregulation of plasma UA concentrations [44].

Our results showed significant downregulation of protein levels of MMP-28 in obese diabetic ZDF rats. This enzyme is the newest identified member of the MMP family and it has been documented that MMP-28 protein levels decrease in response to pathological conditions [45]. In contrast to MMP-2 and MMP-9 where myocardial infarction (MI) induced increase of their levels/activities [46,47], MMP-28 levels decreased post-MI [45]. Moreover, deletion of MMP-28 increased dysfunction of the left ventricle post-MI by reduction of inflammatory and fibrotic responses. In another study, Ma et al. also found that deletion of MMP-28 may accelerate the decline of function of the left ventricle with age [48]. Our data indicate that the observed changes in heart functional parameters due to increased obesity and diabetes development could be associated with the observed decrease of MMP-28. In accordance with this is the fact that quercetin treatment did not modulate negative effects of diabetes on heart functional parameters as well as on MMP-28. Moreover, the application of quercetin also did not influence these parameters in control lean animals. MMP-28 deletion was also found to be associated with decreased collagen deposition [45]. Using a specific antibody, we documented decreased content of collagen I in the left ventricle of obese diabetic ZDF rats. Collagen I is a potential endogenous substrate for MMPs, and the diabetes-induced changes may reflect the observed downregulation of MMP-28 or activation of tissue 72 kDa MMP-2. The observed data show that quercetin did not change collagen I levels in control lean rats and it did not prevent the effects of diabetes.

The present study demonstrated a significant increase in specific Akt kinase phosphorylation in the diabetic myocardium from QCT-treated obese ZDF rats. Several studies demonstrated an important role of PI3K/Akt signaling in mechanisms of heart adaptation to pathological conditions [14,49,50]. However, we found that Akt kinase activation after prolonged QCT application was not associated with positive effects on heart function in diabetic ZDF rats. These findings are different from our previous observations showing positive correlation between QCT-mediated Akt kinase activation and preservation of the left ventricular function in doxorubicin-treated rat hearts [14]. It is interesting to note that treatment with QCT did not change the total Akt level in the left ventricle, but only the active form of Akt. This indicates that QCT-mediated Akt kinase activation did not influence modulation of heart function but may play a role in some other adaptive responses to counteract diabetes-induced negative effects on myocardium. Obesity is characterized by permanently increased oxidative stress [51] and Akt kinase as a part of Akt/Nrf2/ARE signaling plays an important role in increasing the activity of antioxidative enzymes such as SOD and heme oxygenase-1 (HO-1) [52]. In such a way, it can be observed that Akt kinase activation after QCT treatment counteracts diabetes-induced negative effects on SOD activities.

## 4. Materials and Methods

### 4.1. Materials

Quercetin (QCT) was purchased from Sigma Aldrich (cat. No Q4951, St. Louis, MO, USA). Primary antibodies against MMP-28 (ab175937) and collagen I (ab34710) were obtained from Abcam (Cambridge, UK). Antibodies against total Akt kinase (sc-8312), MMP-2 (sc-10736), MMP-9 (sc-393859), TIMP-2 (sc-5539), SOD-1 (sc-17767), SOD-2 (sc-133254), and GAPDH (sc-32233) were obtained from Santa Cruz Biotechnology (Dallas, TX, USA). Primary antibody recognizing phosphorylated Akt kinase (Ser473) (#4058) and secondary peroxidase-labeled anti-rabbit (#7074S) or anti-mouse (#7076S) antibodies were purchased from Cell Signaling Technology (Danvers, MA, USA). The Colorimetric SOD Assay kit (ab65354) was purchased from Abcam (Cambridge, UK).

### 4.2. Experimental Model

In the study, one-year-old (at the beginning of the experiment) obese Zucker Diabetic Fatty (ZDF) rats and their age-matched non-diabetic lean controls were used. All animals were housed at a stable temperature of 22 ± 2 °C and humidity of 45–65%. Rats were fed with normal chow KZ-P/M (complete feed mixture for rats and mouse, reg. no 6147, Dobra Voda, Slovak Republic) and allowed access to drinking water ad libitum. Rats were divided into four experimental groups (*n* = 12–17 in each group: lean vehicle-treated controls (C), lean QCT-treated (Q), obese diabetic vehicle-treated (Dia), and obese diabetic QCT-treated (DiaQ). In the Q and DiaQ groups, the rats received QCT in the dose of 20 mg/kg/day for 6 weeks. QCT was dissolved in a small amount of ethanol and served on a piece of biscuit (vehicle) as described previously [53].

### 4.3. Assessment of Heart Functional Parameters in Langendorff-Perfused Hearts

Heart function was assessed ex vivo using Langendorff method of isolated heart. Briefly, after the completion of QCT or vehicle treatment, animals were anesthetized with thiopental (50 mg/kg, *i.p.*) and heparinized (500 IU, *s.c.*). After rapid thoracotomy, hearts were excised and immediately placed in ice-cold saline, then cannulated *via* the aorta and placed onto Langendorff setup. Hearts were perfused at a constant perfusion pressure of 73 mmHg at 37 °C. Modified Krebs–Henseleit (K–H) buffer gassed with 95% O_2_ and 5% CO_2_ (pH = 7.4) containing (in mmol/L): NaCl 118.0; KCl 3.2; MgSO_4_ 1.2; NaHCO_3_ 25.0; KH_2_PO_4_ 1.18; CaCl_2_ 2.5; glucose 7.0; was used as perfusion solution. K–H buffer was filtered through 1.2 μm porosity filter to remove contaminants before its use for heart perfusion. An epicardial electrocardiogram was registered by the means of two stainless steel electrodes attached to the apex of the heart and aortic cannula. Left ventricular pressure was measured by a water-filled elastic balloon inserted into the left ventricle via the left atrium (adjusted to obtain end-diastolic pressure of 1–5 mmHg) and connected to a pressure transducer (4/30, AD Instruments, Budapest, Hungary). Left ventricular developed pressure (LVDP, systolic minus diastolic pressure), maximal rates of pressure development and fall, +(dP/dt)_max_ and −(dP/dt)_max_ as the indexes of contraction and relaxation, heart rate (HR, calculated from ECG), rate pressure product (RPP, HR x LVDP), and coronary flow were measured to assess cardiac function using PowerLab Chart 7 software (AD Instruments, Budapest, Hungary). Functional parameters of hearts were assessed in the 15th–20th minute of perfusion (after stabilization of all parameters).

### 4.4. Tissue Samples Collection

Six weeks after the completion of QCT or vehicle treatment, the animals were anesthetized with thiopental (50 mg/kg, i.p.) and were euthanized by thoracotomy and rapid excision of their hearts was executed. Excised hearts were weighted and separated to the ventricles. The whole heart and left ventricular weights were registered. Further processing of the collected left ventricular tissue samples was dependent on the following assay. The left ventricular tissue samples for biochemical analysis were immediately frozen in liquid nitrogen and stored at −75 °C until use.

### 4.5. Preparation of Tissue Protein Fractions and Western Blot Analysis

The left ventricular tissue samples were resuspended in ice-cold buffer containing (in mmol/L): 20 Tris-HCl, 250 sucrose, 1.0 dithiothreitol (DTT), 1.0 ethyleneglycol-*bis*(β-aminoethyl)-N,N,Nʹ,Nʹ-tetraacetic acid (EGTA), and 1.0 phenylmethylsulphonyl fluoride (PMSF) (pH 7.4) and were homogenized with a Teflon homogenizer. The homogenates were centrifuged at 800× *g* for 5 min at 4 °C, the pellets were discarded after centrifugation and the supernatants were centrifuged again at 16,100× *g* for 30 min. The supernatants after the second centrifugation were used for biochemical analysis and protein concentrations were estimated by a method of Bradford [54].

For Western blot analysis, samples containing equivalent amounts of proteins per lane were separated by sodium dodecyl sulfate-polyacrylamide gel electrophoresis (SDS-PAGE). The proteins after electrophoretic separation were transferred onto nitrocellulose membranes, and after blocking of non-specific binding sites, the membranes were incubated overnight at 4 °C with the corresponding specific primary antibody. The corresponding peroxidase-labeled anti-rabbit or anti-mouse immunoglobulins (Cell Signaling Technology, Danvers, MA, USA) were used as secondary antibodies. Peroxidase reactions were detected by the enhanced chemiluminescence (ECL) system and quantified using Carestream software (version 5.0, Carestream Health, New Haven, CT, USA).

### 4.6. Measurement of MMPs Activities by Gelatin Zymography

The activities of MMP-2 and MMP-9 were evaluated using zymography in 10% polyacrylamide gels containing gelatin (2 mg/mL) as a substrate. The samples were prepared in Laemmli buffer without 2-mercaptoethanol and loaded onto gels without denaturation. After electrophoresis, the gels were washed twice for 20 min with 50 mmol/L Tris-HCl (pH 7.4), containing 2.5% Triton X-100, and then incubated overnight at 37 °C in a substrate buffer containing 50 mmol/L Tris-HCl, 10 mmol/L CaCl_2_, and 1.25% Triton X-100, pH 7.4. After incubation, the gels were stained with 1% Coomassie Brilliant Blue G-250 and then destained with 40% methanol and 10% acetic acid. The gelatinolytic activities of the MMPs were detected as transparent bands against a dark blue background. Recombinant, active MMP-2 was used as a positive control to identify the active 63 kDa MMP-2 form. Then, 72 kDa MMP-2 and MMP-9 were identified using fetal bovine serum containing predominantly these forms of MMPs.

### 4.7. Determination of Superoxide Dismutase Activity

SOD activities in the left ventricular tissue samples were determined using a colorimetric Superoxide Dismutase Activity Assay Kit (ab65354, Abcam, Cambridge, UK). according to the manufacturer’s protocol. The assay kit utilizes a water soluble tetrazolium salt WST-1 that produces a formazan dye upon reduction with superoxide anion. The greater the activity of SOD in the sample, the less formazan dye is produced. The formazan production was determined by measurement of absorbance at 450 nm.

### 4.8. Statistical Evaluation

Distribution of variables was examined using Shapiro–Wilk test of normality and the data from measurements were further analyzed by the unpaired Student’s *t*-test. Differences were considered significant at *p* < 0.05 in all the tests. All analyses were carried out with GraphPad Prism version 6.0c (GraphPad Software, San Diego, CA, USA).

## 5. Conclusions

Taken together, obtained results demonstrate that increased obesity has a negative impact on hemodynamic parameters of the hearts in one-year-old diabetic ZDF rats. The findings of the current study also demonstrate that diabetes development involves changes in MMP-2 and MMP-28 and attenuation of antioxidant defense by SOD. Prolonged treatment with QCT did not have positive effects on improvement of heart function or modulation of MMPs in diabetic hearts. Nevertheless, application of QCT mediated activation of adaptive myocardial responses against oxidative stress through Akt kinase and prevention of diabetes-induced negative effects on SOD activities.

## Figures and Tables

**Figure 1 ijms-22-04457-f001:**
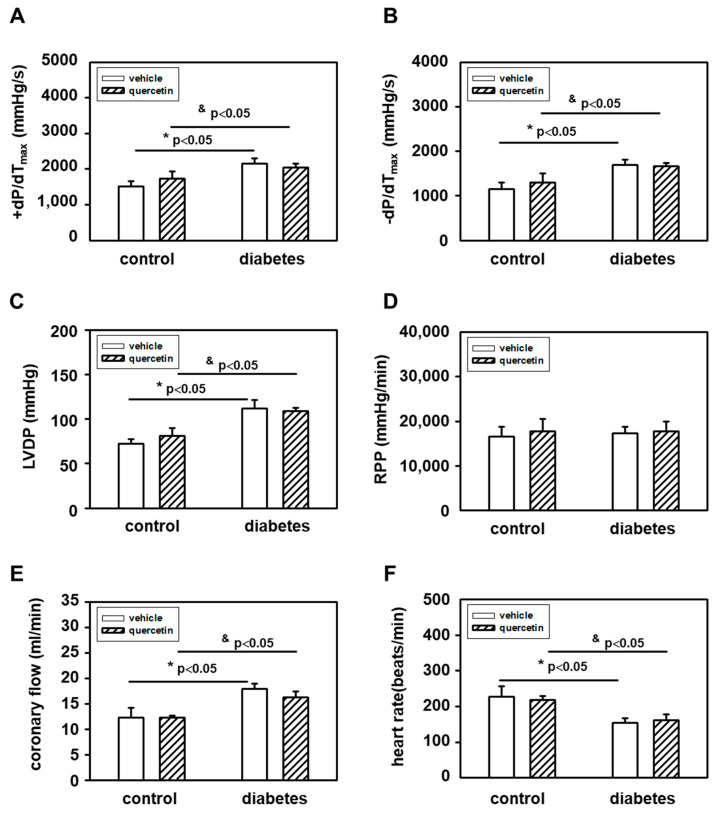
Effects of quercetin on functional parameters of hearts measured ex vivo on Langendorff perfusion setup: (**A**) +(dP(dt)_max_; (**B**) −(dp/dt)_max_; (**C**) left ventricle developed pressure; (**D**) rate-pressure product; (**E**) coronary flow; and (**F**) heart rate. Diabetes significantly increased −(dp/dt)_max_, LVDP, and coronary flow, and significantly decreased heart rate. Quercetin had no effect on functional parameters of hearts. Abbreviations: LVDP—left ventricle developed pressure; RPP—rate-pressure product (LVDP x heart rate); ± (dP/dt)_max_—maximal rates of pressure development/fall. Data are presented as means ± SEM, *n* = 6–7. Statistical significance was analyzed by unpaired *t*-test. * *p <* 0.05 vs. control lean vehicle-treated rats, ^&^
*p <* 0.05 vs. control lean quercetin-treated rats.

**Figure 2 ijms-22-04457-f002:**
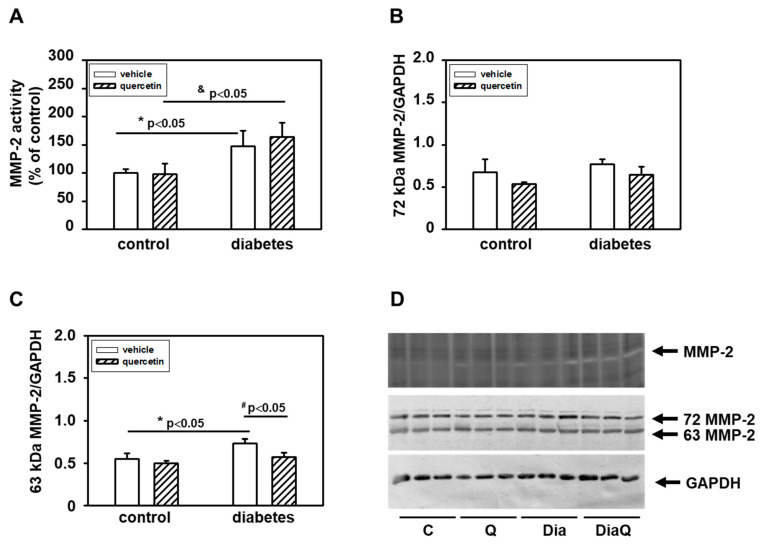
(**A**) Quantitative analysis of the tissue 72-kDa matrix metalloproteinase-2 (MMP-2) activities. Activities were analyzed using gelatin zymography and data are expressed as a percentage of value for corresponding control. Each bar represents mean ± SEM (*n* = 6–7) and statistical significance was revealed by Student’s unpaired *t*-test, * *p <* 0.05 vs. control lean vehicle-treated rats, ^&^
*p <* 0.05 vs. control lean quercetin-treated rats. (**B**) Quantification of 72-kDa MMP-2 content normalized to the GAPDH protein levels. (**C**) Quantification of 63-kDa MMP-2 content normalized to the glyceraldehyde-3-phosphate dehydrogenase (GAPDH) protein levels. Data were obtained from Western blot records and each bar represents mean ± SEM, *n* = 6–7. Statistical significance was revealed by Student’s unpaired *t*-test, * *p <* 0.05 vs. control lean vehicle-treated rats, ^#^
*p <* 0.05 vs. obese diabetic vehicle-treated rats. (**D**) At the top is a record showing the activities of MMP-2 analyzed using gelatin zymography; in the middle, a Western blot record showing MMP-2 protein levels analyzed using a specific antibody that reacts with both the 72 and 63 kDa forms of MMP-2, and at the bottom the protein loading using GAPDH is documented. C—control lean vehicle-treated ZDF rats; Q—lean quercetin-treated ZDF rats; Dia—obese diabetic vehicle-treated ZDF rats; DiaQ—obese diabetic quercetin-treated ZDF rats.

**Figure 3 ijms-22-04457-f003:**
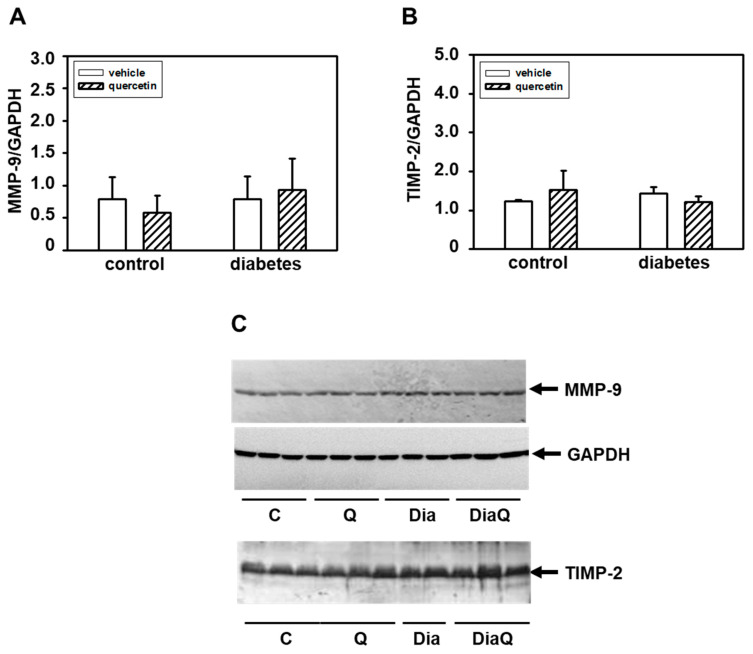
Effects of quercetin treatment on protein levels of matrix metalloproteinase-9 (MMP-9) and matrix metalloproteinases tissue inhibitor-2 (TIMP-2) in control lean and obese diabetic Zucker Diabetic Fatty (ZDF) rats. (**A**) Quantitative analysis of the MMP-9. Protein levels of MMP-9 in the left ventricle were determined by Western blot analysis using a specific antibody and quantifications of MMP-9 content were normalized to the glyceraldehyde-3-phosphate dehydrogenase (GAPDH) protein levels. Each bar represents mean ± SEM and statistical significance was revealed by Student’s unpaired *t*-test. (**B**) Quantitative analysis of the TIMP-2. Protein levels of TIMP-2 in the left ventricle were determined by Western blot analysis using a specific antibody and quantifications of TIMP-2 content were normalized to the GAPDH protein levels. Each bar represents mean ± SEM, *n* = 6–7. Statistical significance was revealed by Student’s unpaired *t*-test. (**C**) Western blot records showing the protein levels of MMP-9, TIMP-2, and GAPDH. C—control lean vehicle-treated ZDF rats; Q—lean quercetin-treated ZDF rats; Dia—obese diabetic vehicle-treated ZDF rats; DiaQ—obese diabetic quercetin-treated ZDF rats.

**Figure 4 ijms-22-04457-f004:**
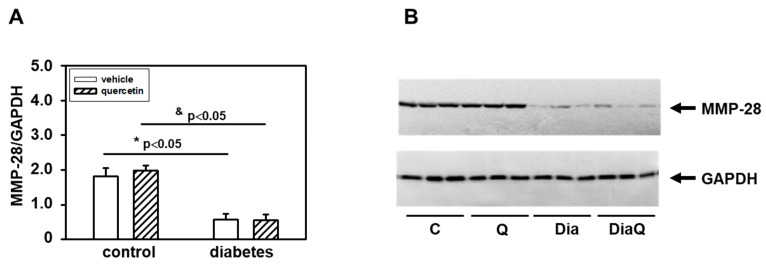
Effects of quercetin treatment on protein levels of matrix metalloproteinase-28 (MMP-28) in left ventricular tissue of control lean and obese diabetic Zucker Diabetic Fatty (ZDF) rats. (**A**) Quantification of MMP-28 content normalized to glyceraldehyde-3-phosphate dehydrogenase (GAPDH) protein levels. Each bar represents mean ± SEM, *n* = 6–7. Statistical significance was revealed by Student’s unpaired *t*-test, * *p <* 0.05 vs. control lean vehicle-treated rats, ^&^
*p <* 0.05 vs. control lean quercetin-treated rats. (**B**) Western blots records showing the changes in protein levels of MMP-28 and GAPDH. C—control lean vehicle-treated ZDF rats; Q—lean quercetin-treated ZDF rats; Dia—obese diabetic vehicle-treated ZDF rats; DiaQ—obese diabetic quercetin-treated ZDF rats.

**Figure 5 ijms-22-04457-f005:**
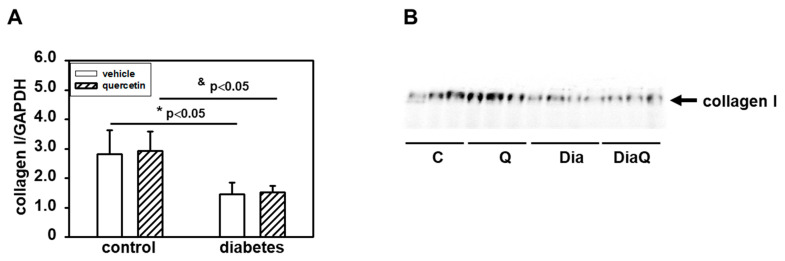
Effects of quercetin treatment on collagen I content in left ventricular tissue of control lean and obese diabetic Zucker Diabetic Fatty (ZDF) rats. (**A**) Collagen I levels were determined by Western blot analysis using a specific antibody. Quantification of collagen I content was normalized to glyceraldehyde-3-phosphate dehydrogenase (GAPDH) protein levels. Each bar represents mean ± SEM, *n* = 6–7. Statistical significance was revealed by Student’s unpaired *t*-test, * *p <* 0.05 vs. control lean vehicle-treated rats, ^&^
*p <* 0.05 vs. control lean quercetin-treated rats. (**B**) Western blot records showing the changes in collagen I levels. C—control lean vehicle-treated ZDF rats; Q—lean quercetin-treated ZDF rats; Dia—obese diabetic vehicle-treated ZDF rats; DiaQ—obese diabetic quercetin-treated ZDF rats.

**Figure 6 ijms-22-04457-f006:**
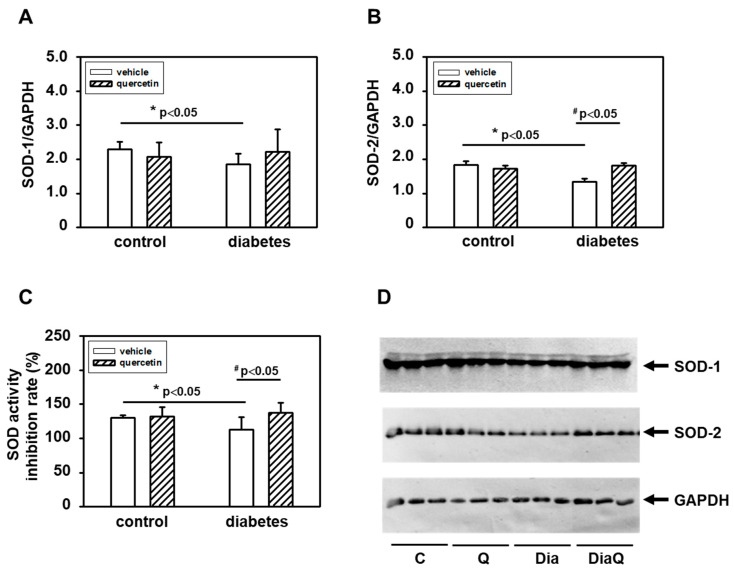
Effects of quercetin on superoxide dismutase (SOD) protein levels and activities in left ventricular tissue of control lean and obese diabetic Zucker Diabetic Fatty (ZDF) rats. (**A**) Quantification of SOD-1 protein levels. (**B**) Quantification of SOD-2 protein levels. Levels of SOD-1 and SOD-2 were determined by Western blot analysis using specific antibodies. Quantifications of SOD-1 and SOD-2 content were normalized to the glyceraldehyde 3-phosphate dehydrogenase (GAPDH) protein levels. Each bar represents mean ± SEM, *n* = 6–7. Statistical significance was revealed by Student’s unpaired *t*-test, * *p <* 0.05 vs. control lean vehicle-treated rats, ^#^
*p <* 0.05 vs. obese diabetic vehicle-treated rats. (**C**) Effects of quercetin on SOD activities. SOD activities were determined using SOD colorimetric assay kit (Abcam) in cytosolic fractions isolated from the left ventricular tissue (*n* = 6–7). The data represent the percentage inhibition of superoxide production by SOD activity. (**D**) Western blot records showing the changes in protein levels of SOD-1, SOD-2, and GAPDH. C—control lean vehicle-treated ZDF rats; Q—lean quercetin-treated ZDF rats; Dia—obese diabetic vehicle-treated ZDF rats; DiaQ—obese diabetic quercetin-treated ZDF rats.

**Figure 7 ijms-22-04457-f007:**
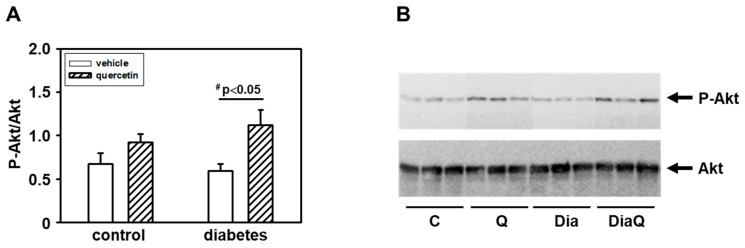
Effects of quercetin treatment on Akt kinase activation in left ventricular tissue of control lean and obese diabetic Zucker Diabetic Fatty (ZDF) rats. (**A**) The quantification of Akt kinase phosphorylation (activation). Data were obtained from Western blot records and are expressed as a ratio of content of serine 473(Ser473) phosphorylated Akt kinase to total Akt kinase. Each bar represents mean ± SEM, *n* = 6–7. Statistical significance was revealed by Student’s unpaired *t*-test, ^#^
*p <* 0.05 vs. obese diabetic vehicle-treated rats. (**B**) Western blot records showing the changes in protein levels and specific phosphorylation of Akt kinase. The changes in activation of Akt kinase were determined using an antibody which reacts with Akt kinase phosphorylated specifically on Ser473. C—control lean vehicle-treated ZDF rats; Q—lean quercetin-treated ZDF rats; Dia—obese diabetic vehicle-treated ZDF rats; DiaQ—obese diabetic quercetin-treated ZDF rats.

**Table 1 ijms-22-04457-t001:** Effects of vehicle and quercetin treatment on biometric and biochemical characteristics in control lean and obese diabetic one-year-old Zucker Diabetic Fatty (ZDF) rats. Abbreviations: C, lean vehicle-treated controls; Q, quercetin-treated lean rats; Dia, vehicle-treated obese diabetic rats; DiaQ, quercetin-treated obese diabetic rats; BW, body weight; TAG, triacylglycerides; HDL, high-density lipoprotein; LDL, low-density lipoprotein. Data are presented as means ± SEM, *n =* 12–14. Significant differences were evaluated by *t*-test, * *p <* 0.05 vs. control lean vehicle-treated rats, ^&^
*p <* 0.05 vs. control lean quercetin-treated rats.

Parameter	C	Q	Dia	DiaQ
BW (g)	406 ± 10	413 ± 8	520 ± 25 *	514 ± 24 ^&^
Glucose (mmol/L)	6.1 ± 0.1	5.9 ± 0.2	16.9 ± 1.6 *	18.48 ± 2.1 ^&^
TAG (mmol/L)	0.24 ± 0.05	0.18 ± 0.03	3.66 ± 0.41 *	3.67 ± 0.39 ^&^
Cholesterol (mmol/L)	2.85 ± 0.11	2.81 ± 0.08	4.86 ± 0.41 *	4.79 ± 0.25 ^&^
HDL-cholesterol (mmol/L)	1.36 ± 0.04	1.37 ± 0.04	2.36 ± 0.13 *	2.37 ± 0.07 ^&^
LDL-cholesterol (mmol/L)	0.82 ± 0.04	0.81 ± 0.02	1.05 ± 0.08 *	0.91 ± 0.07 ^&^

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
