# Peer review of "Matrix Metalloproteinases and Their Role in Mechanisms Underlying Effects of Quercetin on Heart Function in Aged Zucker Diabetic Fatty Rats"

_ijms, 2021, doi:10.3390/ijms22094457_

Round 1

Reviewer 1 Report

In this manuscript the authors have demonstrated the effects of  QCT administration on changes in heart function in aged lean and obese Zucker Diabetic Fatty (ZDF) rats and that in association to MMPs. the study looks very clean and straight forward. However, there are few concerns and comments the authors need to address:

  1. Figure 1: Put a line on top of the bars in the bar graphs denoting which groups are being compared and keep the actual p value on top of the lines. Mention the n numbers in the figure legend.
  2. Figure 2: As mentioned in the comment 1, similarly, please put p values/n numbers and put lines on top of the bars. Please quantify the band intensity of Fig. 2D and represent as a bar graph and do a t-test for better comparison.
  3. Figure 3: Put p values on top of the bar graph (Fig. 3A) and n number in the figure legends. The authors are requested to provide a comparatively cleaner blot of TIMP-2 panel. It is very difficult to compare the band intensity of TIMP-2 in different groups. If possible represent the blots as bar graphs and do a t-test.
  4. Figure 4: For Fig 4A put the p values and represent the western blot as bar graph (Fig 4B) as mentioned for the previous figures.
  5. Figure 5: For Fig 5A put the p values as mentioned for the previous figures. The authors are requested to provide a cleaner blot of collagen I and represent the western blot as bar graph with p values (Fig 5B).
  6. Figure 6 and 7: Put the p values in the bar graphs as mentioned previously and represent the western blot as bar graph with p values.

Reviewer 2 Report

I’ve read with attention the paper of Boťanská et al. that is potentially of interest. The background and aim of the study have been clearly defined. The methodology applied is overall correct, the results are reliable and adequately discussed. I’ve only some minor comments: - The authors have failed to consider the effect of Quercetin on serum uric acid, that per se could impact heart function. This has to be mentioned and shortly discussed

- I strongly doubt that all the tested parameters had normal distribution with a so small sample size, so Student t-test could not be applied to all the analyses.

Round 2

Reviewer 2 Report

The authors have considered the reviewers’ suggestion and improved the paper accordingly. I’ve no further comments on it.